# A Modified Protocol of Diethylnitrosamine Administration in Mice to Model Hepatocellular Carcinoma

**DOI:** 10.3390/ijms21155461

**Published:** 2020-07-30

**Authors:** Azra Memon, Yuliya Pyao, Yerin Jung, Jung Il Lee, Woon Kyu Lee

**Affiliations:** 1Department of Biomedical Sciences, School of Medicine, Inha University, Incheon 22212, Korea; azrabiochem@yahoo.com (A.M.); julipyao@gmail.com (Y.P.); 22191327@inha.edu (Y.J.); 2Department of Internal Medicine, Gangnam Severance Hospital, Yonsei University College of Medicine, Seoul 06273, Korea; MDFLORENCE@yuhs.ac

**Keywords:** animal model, HCC, DEN, cirrhosis

## Abstract

We aimed to create an animal model for hepatocellular carcinoma (HCC) with a short time, a high survival rate, as well as a high incidence of HCC in both males and females than previously reported. The Diethylnitrosamine (DEN) model has an age-related effect. A single dose of DEN treatment is not enough in young mice up to 50 weeks. The same pattern is shown in an adult with multiple-dose trials whether or not there is some promotion agent. In this study, two-week old C57BL6 mice were given a total of eight doses of DEN, initially 20mg/kg body weight, and then 30mg/kg in the third week, followed by 50mg/kg for the last six weeks. The first group is DEN treatment only and the other two groups received thioacetamide (TAA) treatment for four or eight weeks after one week of rest from the last DEN treatment. An autopsy was performed after 24 weeks of the initial dose of DEN in each group. The cellular arrangement of HCC in the entire group was well-differentiated carcinoma and tumor presence with no significant impact on the survival of mice. Increased levels of the biochemical markers in serum, loss of tissue architecture, hepatocyte death, and proliferation were highly activated in all tumor-induced groups. This finding demonstrates an improved strategy to generate an animal model with a high occurrence of tumors combined with cirrhosis in a short time regardless of sex for researchers who want to investigate liver cancer-related.

## 1. Introduction

Hepatocellular carcinoma (HCC) is one of the most aggressive clinical conditions with high malignancy, quick progression, and poor prognosis [1,2]. Hepatocellular carcinoma is a chronic liver disease that starts as fibrosis, leads to cirrhosis, and results in cancer. The treatment of hepatocellular carcinoma is often complex as cirrhosis is the major clinical risk factor associated with 80% of people with HCC [3], and the potential for surgical resection is limited because of the risk of inducing postoperative liver failure. Liver transplantation (LT) offers the best chance of long-term survival, but cannot be offered to all patients because of a shortage of organs [4]. The pathophysiological background of HCC is still poorly understood. Therefore, it is necessary to identify ideal experimental HCC models with or without cirrhosis features similar to human HCC. In vivo, experimental models are playing a significant role in the study of HCC [5]. Genetic, transgenic, or conditional knock-out experimental models are mainly used in the study of HCC development [6,7]. Numerous chemicals promote HCC in rodents, of which DEN is most often used to promote cancer development. The application of DEN at different concentrations and to animals of different ages has variant efficacy and efficiency.

The simple and effective protocol is a single injection of DEN dosing from 1.25 to 25 mg/kg body weight using two-week old mice a time at which hepatocytes are still proliferative, resulting in the appearance of HCC at around 8–12 months [8]. The advantage of chemically induced models, especially the DEN-induced HCC mouse model, is the resemblance of gene expression patterns to human HCC poor prognosis class [9,10]. Both DEN-induced tumors and human HCC with poor prognosis have high proliferation, ubiquitination, and chromosomal instability, as well as low levels of β-catenin mutation and apoptosis [11]. DEN administration can induce clinical HCC indicators GPC3 and alpha-foetoprotein AFP expression [12,13]. DEN is bioactivated by cytochrome P450, leading to the formation of mutagenic DNA adducts while also generating reactive oxygen species (ROS) which damage DNA, oxidative stress, and finally promote hepatocarcinogenesis [14,15,16]. These protocols are widely used in the literature and represent a good model to define and investigate primitive HCC nodules independently from the condition of cirrhosis. Still, the DEN model requires other hallmarks of human HCC, especially fibrosis in the surrounding microenvironment [17].

Many studies have investigated the development of HCC animal models to reduce the time and death ratio and to limit the administration of carcinogens. Moreover, reports commonly use a two-stage model in which a single dose of DEN followed by phenobarbital (PB) is applied as a promoting agent [18,19,20,21]. Although this influences DEN-initiated mice, the effects vary considerably depending upon gender, age, and strain [8,22,23]. Another two-step hepatocarcinogenesis model initiated by a hepatocarcinogenic compound followed by partial hepatectomy (PH) induces hepatic cell proliferation of the liver, which leads to the rapid development of the initiated cells as well as the rapid occurrence of altered hepatic foci and visible nodules [24,25].

A two-stage model in which the initiation by a genotoxic compound DEN followed by a promotion phase with non-genotoxic thioacetamide (TAA) is often used for inducing HCC [26]. TAA is bioactivated by cytochrome P450 enzymes to TAASO and further to TAASO2 with a toxic nature that elevates levels of reactive oxygen species (ROS), lipid peroxides, cytotoxicity, mitochondrial injury, and glutathione depletion. Finally, this leads to centrilobular necrosis, apoptosis, and the formation of hepatocellular carcinoma and cholangiocarcinoma [27]. The treatment with a tumor-promoting agent like TAA has been shown to induce liver fibrosis development in several different mouse strains, different doses, administration routes, and treatment durations [26,28,29,30,31,32,33,34]. TAA-promotion after initiation with DEN increased the number and area of hepatocellular foci, numbers of proliferating and apoptotic cells [26]. HCC is a multistep process involving repeated rounds of treatment that lead to inflammation, fibrosis, and eventually HCC.

One of the issues to make the HCC animal model is the production period and survival rate. It takes 8–12 months to show carcinogenesis with a high death ratio [35,36,37,38]. This is why the appropriate period is important to make this model mirror the diversity of causes seen in patients such as steatosis, inflammation, fibrosis, and cirrhosis.

Based on current literature, genotoxic agents like DEN directly induce tumor formation and, when combined with promoting agents like TAA, enhance both tumor development and its aggressiveness with fibrosis in the surrounding microenvironment. Taking into consideration the reproducibility of HCC development, the time required and the survival rate of the animal models are at potential risk to the investigator. Appropriate animal models are essential to the investigation of HCC with or without a cirrhotic background and to providing information regarding its molecular, cellular, and pathophysiological mechanisms, as well as to the development and challenge for the novel therapeutic processes. Therefore, the present study was conducted to develop a detailed protocol that appears to be a suitable animal model for HCC after cirrhosis with a short modeling time, improved survival rate, and increased tumor incidence for researchers who want to use liver cancer animal models.

## 2. Results

### 2.1. General Observations

Throughout the experimental period, body weight in the control group progressively increased every time whereas body weight in all treatment groups increased slowly at first, then decreased after the last injection of DEN which was again slowly recovered after two weeks of the last injection. During this whole experiment period, one mouse from the DEN and two mice from the DEN+TAA for 8 weeks treatment group died. The mice whose body weight was more than 5 g at the first DEN injection time survived until the end of the experiment while mice less than 5 g in body weight died during the experiment (Figure 1b). As shown in Figure 1c and Table 1, the final body weights were significantly decreased in all treated groups relative to the mock control group (*p* < 0.01), while the liver weights in the treated groups increased (*p* < 0.01, Figure 1d), which is a common feature of HCC [37,39]. The survival ratio was 97% in the DEN group and 100% in the D+T 4 wks group, while the D+T 8 wks group showed 85% animal survival until the end of the experiment (Figure 1b).

### 2.2. Liver Tumors

Immediately after sacrifice, the livers were removed, weighed, and the numbers of visible tumors on the liver surface were counted macroscopically. All animals either male or female treated with either DEN alone or DEN+TAA for four or eight weeks developed tumors, as shown in Figure 1e and Appendix A, but the size decreased in the TAA combination group when compared to the DEN single treatment, which may have been correlated with cirrhosis progression during tumor development. The overall tumor number was nearly similar in the DEN and D+T 4 wks group, while it increased in the D+T 8 wks group (Table 2, Appendix A).

### 2.3. Assessment of HCC

The presence of HCC was confirmed by histopathologic examination of the liver sections with H&E staining. The mock control group showed the normal architecture, whereas the DEN treated group showed loss of architecture and the presence of tumors (Figure 2a). Some non-tumor areas also showing numerous huge pathological vacuoles in hepatocytes suggest those lipid droplets accumulated in the liver and that these animals developed hepatic steatosis (Appendix Aa). Liver nodules presented as basophilic foci with crowded nuclei and were classified as a typical HCC (Appendix Ab). Adenomas were distinguished from atypical foci based on the presence of clearly defined margins.

### 2.4. Biochemical Markers

The release of alanine aminotransferase and aspartate aminotransferase in serum analyzed which was used as an indicator of liver injury. The mean serum ALT and AST levels in the DEN-treated group were three-fold greater than untreated or mock control mice. Mice treated with combination doses of TAA for further four or eight weeks showed a gradual elevation of ALT and AST serum concentration respectively. ALT and AST concentrations were statistically significant at all three treated groups (*p* < 0.01) (Figure 2b,c).

### 2.5. Hepatocellular Death and Proliferation

We measured tumor cell proliferation by determining the number of hepatic tumor cells that were positively immunostained for PCNA in the livers of mice from different groups (Figure 3a,b). We counted positive stains for proliferating cells, which showed a highly-increased amount of proliferated cells in all groups relative to mock control mice (p < 0.01). The PCNA positive cells were scored blind, and 400–700 hepatocytes per field were counted from 6–8 random fields of each mouse liver with a total of three mice per group. Moreover, the TUNEL apoptosis assay revealed more apoptotic cells in treated groups relative to the controls (Figure 3b,d). These results suggest that increased proliferation and cell death occurred in liver tumors during HCC progression.

### 2.6. Fibrogenesis

Collagen accumulation is a hallmark of liver fibrogenesis. We first assessed the collagen deposition by Sirius red staining. Representative images and quantification of Sirius red staining of liver sections revealed that little collagen was deposited in the livers of control mice or DEN-treated mice, but that TAA treated mice contained high levels of collagen, especially in the D+T 8 weeks group (Figure 4a,b). As expected, the protein expression of collagen 1 (Col1) was generally enhanced in fibrotic/cirrhotic mice, but the pattern and degree of changes varied among groups (Figure 4c,e). Consistent with the results of Sirius red staining, a significant increase in collagen 1 was found in groups treated with additional TAA for four weeks, with peak levels occurring in animals in the eight-week TAA treatment group (*p* < 0.01). However, significant differences were observed between the control and DEN treated groups (Figure 4c,e). Cyclooxygenase (COX)-2, which is also used as an HCC marker, is upregulated in hepatocellular carcinoma (HCC). Cox2 expression enhanced in DEN treated all groups as compared to the control group (Figure 4c,d).

## 3. Discussion

Several animal models are commonly used to understand the molecular and cellular basis of HCC, and DEN is a very common carcinogen used to induce HCC in rodents. The main limitation of the application of DEN to generate HCC models is the long duration of the experiments. In mice, most studies employing various methods to accelerate hepatocarcinogenesis have shown that the average time required for tumor formation is still quite long and tumor incidence is low. Previous studies have demonstrated that adult mice treated with DEN did not form tumors for one year [40]. In adult mice, DEN is a weak carcinogen that requires promoting agents such as PB, which may induce a higher rate of carcinogenesis, but could also modify the characteristics of the tumor slightly as well as reduce the reproducibility of the model [41,42]. Additionally, the PH-method is based on a difficult surgical technique and is therefore operator-dependent and less reproducible [24]. Previous findings suggested that a single DEN dose to infant mice results in efficient HCC induction. Shu et al. reported that DEN exposure in mice with a C57BL/6 background at the age of 15 days showed 21.4% tumor incidence at 35 weeks [38]. Another study reported that 45% of mice developed HCC in response to DEN at 36 weeks [43,44]. Moreover, low-dose (5 µg/g/BW) DEN can cause hepatocellular adenoma at 44 weeks and hepatocellular carcinoma at 68 weeks, with the sequential emergence of preneoplastic hepatic foci, HCAs, and then HCC [8]. HCC incidence in response to DEN treatment occurs 80–100% in mice till 105 weeks [45]. Therefore, the present study was conducted to address this gap by generating a new animal model with a high tumor incidence and low mortality rate that could be utilized within a short period.

In the long-duration experiment, almost all DEN injected mice died by the time they reached 80 weeks of age, even though they were treated at the age of 14 days with a single DEN dose of 25 mg/kg body weight [38]. In adult mice, a single dose of DEN (200 mg/kg) with combined treatment of TAA (200mg/kg) for four weeks resulted in a very low survival rate of 20% [37]. In the present investigation, the survival rate was more than 85% in the entire group until the end of the experiment. Furthermore, based on our regular two checks every week, there were no clinical signs including feeding, appearance, behavior, and growth until 24 weeks. The model didn’t have other abnormalities except for liver cancer at necropsy.

In previous studies of DEN-induced mouse liver cancer, females showed fewer tumors than males [46], and only 8% of females developed HCC, and males produced 1.6 numbers per mouse [47]. However, in the present study, there were 7.4 tumors per mouse (SD 3.42), and all-female developed HCC in all treated groups. This may have occurred because DEN doses were administered at an early age. Interestingly, the mice treated with DEN alone showed significantly larger and greater numbers of tumors, while treatment with DEN+TAA for four or eight weeks resulted in many small tumors with a cirrhotic background. H&E staining confirmed microscopic tumor foci in all three groups. Furthermore, the activity of serum ALT and AST enzymes leakage into the bloodstream indicate the severity of hepatic damage, [48,49] which is also confirmed in all our treated groups as compared to untreated control group results.

Recent studies suggest that COX-2 is chronically over-expressed in HCC [50,51,52]. Clinical studies indicate that it is an important molecular target for anticancer therapies, and COX-2 inhibitors are seen to have anticancer effects in HCC [41,42]. In transgenic mice, the overexpression of COX-2 was adequate to induce tumorigenesis [51]. Our study suggests a high level of COX-2 expression in the DEN group and much more in the DEN+TAA group dose-dependently showing its possible application as an HCC animal model.

Previous studies have indicated that hepatocyte proliferation and cell death in response to DEN exposure is associated with apoptosis-induced compensatory proliferation and HCC development [36,46,53,54]. Similarly, DEN and DEN+TAA treated groups showed a higher number of proliferating cells, which greater increases observed in the groups treated with DEN+TAA for four and eight weeks. These results are consistent with those of a human subclass of HCC with poor prognosis, in which proapoptotic markers were downregulated in response to the upregulation of proliferating markers in tumors [55]. Overall, the present study provides strong evidence that DEN alone can cause HCC with a low mortality rate and high tumor incidence in a reduced time compared to conventional models.

## 4. Materials and Methods

### 4.1. Chemicals

DEN purchased from Sigma-Aldrich (cat# N0756, St. Louis, MO, USA) was freshly dissolved in 0.9% saline and used at various concentrations. Thioacetamide (TAA) from Sigma-Aldrich (cat #172502 St. Louis, MO, USA) was freshly dissolved in phosphate-buffered saline (PBS) before use.

### 4.2. Animal

In this study, 14.5-day-old C57BL/6 mice were used. The number of total mice in each group is given in Table 2. All mice were maintained in filter-topped cages on an irradiated regular chow diet at 22 ± 2 °C and 50–60% relative humidity under 12-h light-dark cycles throughout the experiment. Animal care and all experimental procedures were conducted under the approval and guidelines of the INHA Institutional Animal Care and Use Committee (INHA IACUC) of the medical school of INHA University (Approval ID: INHA 160901-435).

### 4.3. Experimental Design

The mouse body weight of the mice was determined at the time of dosing for the chemical introduced. A total eight doses of DEN was given in our experiment, the first start dose of DEN was 20 mg/kg body weight, the second dose was 30 mg/kg body and the next six doses were 50 mg/kg body weight, all of which were injected intraperitoneally (Figure 1a). After one week of rest, an additional 4 or 8 weeks of TAA treatment was administered for the cirrhosis model. Specifically, TAA (300 mg/kg body weight) was injected twice a week intraperitoneally (Figure 1a). At 3–4 weeks of age, animals were weaned from their mothers and body weight recorded weekly. Mice were sacrificed at 24 weeks after the first dose, after which the sera and livers were collected for further analysis

### 4.4. Histopathological Examination

Immediately after sacrifice, externally visible tumors larger than 0.5 mm were counted by stereo microscopy. The right lobe was then fixed in 10% formalin and processed for histological examination by H&E staining, after which it was further examined microscopically for evidence of tumors that could not be visualized externally.

### 4.5. Blood Chemistry

Blood was collected by heart puncture at the end of the experiment. Samples were kept at room temperature for 30 min and then centrifuged at 3000 rpm for 10 min at 4 °C, after which the serum was kept at −80 °C until use. Alanine transaminase (ALT) and aspartate aminotransferase (AST) were determined using the standard protocol.

### 4.6. Immunohistochemistry

Anti-Proliferative cell nuclear antigen PCNA (sc-56, Santa Cruz, CA, USA) was used for the quantification of compensatory proliferation on liver section slides, in which at least nine microscope fields at 40× magnification (3 per group) were examined. Hepatocyte apoptosis was detected using a TUNEL (TdT-mediated dUTP nick end labeling) ApopTag^®^ Peroxidase In situ Apoptosis detection kit (Cat# S7100 Merk Millipore, Temecula, CA) following the manufacturer’s instructions. The positive signals of IHC were measured using the SPOT Software (Version 4.6, Flex 64 Mp mosaic camera).

### 4.7. Quantification of Fibrosis

Liver sections were stained using a 0.1% PicroSirius red stain kit (ab150681, Abcam Cambridge, UK) according to the manufacturer’s instructions. The relative fibrotic area, expressed as a percentage of total liver area, was assessed by analyzing nine fields of the slide per animal. Each field was acquired at 20× magnification with the SPOT Software (Version 4.6, Flex 64 Mp mosaic camera). The results were analyzed using imaging software (Image J, NIH). To evaluate the relative data, the measured fibrosis area was divided by the net field area and then multiplied by 100.

### 4.8. Protein Extraction and Western Blot Analysis

Total protein was extracted from the liver (100 mg) using 500 μL lysis buffer [50 mM Tris–HCl (pH 8.0), 150 mM NaCl, 0.1% sodium dodecyl sulfate (SDS), 0.5% deoxycholic acid, and 1% NP-40 containing protease inhibitors (Protease Inhibitor Cocktail Set I, Calbiochem, San Diego, CA, USA). The concentration of proteins was determined by bicinchoninic acid (BCA) protein assay (Pierce, Thermo Scientific, Rockford, IL, USA) according to the manufacturer’s instructions. Proteins (20 μg/lane) were separated by SDS–polyacrylamide gel electrophoresis. After electrophoresis, the proteins were electrotransferred to polyvinylidene fluoride (PVDF) membranes, blocked in 5% non-fat milk for 1 h at room temperature (RT), and washed with TBST (TBS with 0.1% Tween20), after which they were probed with anti-col1 (ab63080, Abcam) and Purified Mouse anti-Cox-2 (610203, Becton, Dickinson and Company, Franklin Lakes, NJ, USA). For these analyses, β-actin (ab8224) served as a loading control from Abcam (Abcam, Cambridge, UK). Membranes were incubated with appropriate horseradish peroxidase-conjugated secondary antibody (Cat# sc-2005 Santa Cruz, CA, USA). Each membrane was developed using an enhanced chemiluminescent substrate for the detection of horseradish peroxidase (Cat# 32209 Pierce, Thermo Scientific, Rockford, IL, USA), followed by densitometric scanning using the NIH Image J software version 1.45.

### 4.9. Data Analysis and Statistics

Statistical analysis was performed using the GraphPad Prism 5 software. Data are expressed as Means ± SD. A *p* < 0.05 was considered significant.

## 5. Conclusion

In conclusion, infant C57BL/6 mice treated with continuous dosing of DEN had a high tumor incidence after only six months in both genders with an improved survival ratio. Also, the HCC model of combined liver cirrhosis was successfully established. These models will be useful for future studies in the field of liver cancer.

## 6. Patent

Method for preparing hepatocellular carcinoma with liver cirrhosis animal model. Woon Kyu Lee and Azra Memon, Korea Patent 10-2117590 (2020).

## Figures and Tables

**Figure 1 ijms-21-05461-f001:**
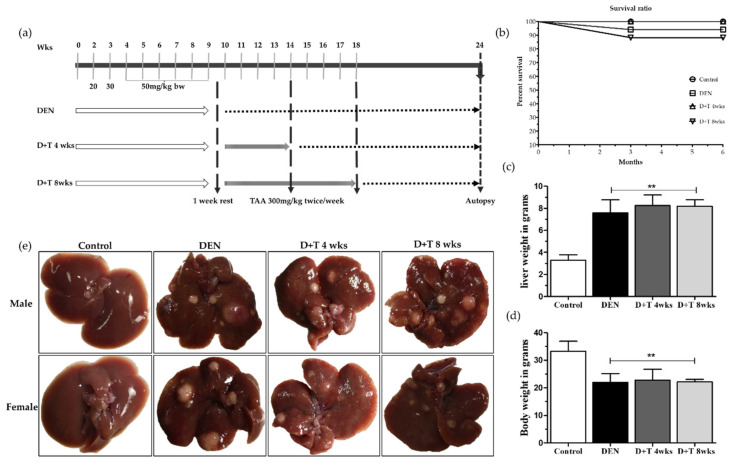
A tumor development pattern in the animal model. (**a**) Grouping and schedule. (**b**) The survival ratio in all four groups at the end of the experiment. (**c**) Final body weight for all groups. (**d**) Relative liver weight for all groups. (**e**) Macroscopic view of representative images of mice livers 6 months after DEN administration. Data are presented as means ± SD. ** Significantly different from the control group (*p* < 0.01).

**Figure 2 ijms-21-05461-f002:**
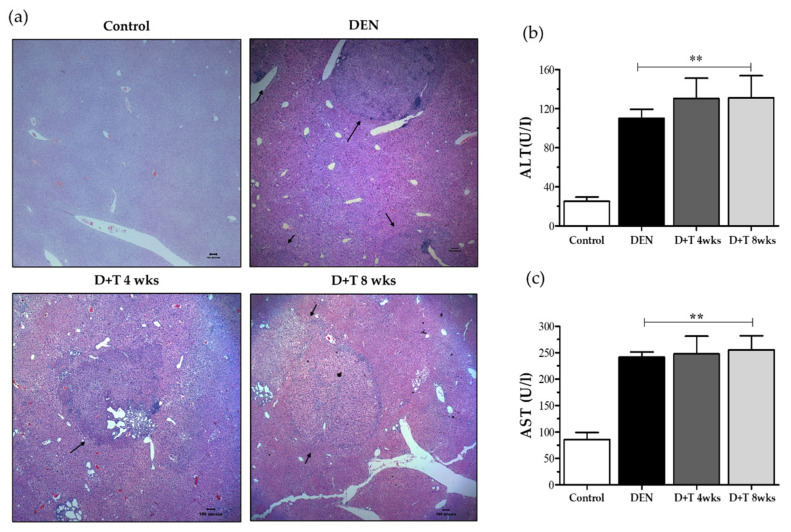
Liver damage pattern. (**a**) Representative images of H&E staining from all groups after 6 months of DEN treatment (40×). Black arrows indicate tumors. (**b**) AST level in serum. (**c**) ALT level in serum. values are given as the means ± SD. At least three mice in each group were recorded. ** Significantly different from the control group (*p* < 0.01).

**Figure 3 ijms-21-05461-f003:**
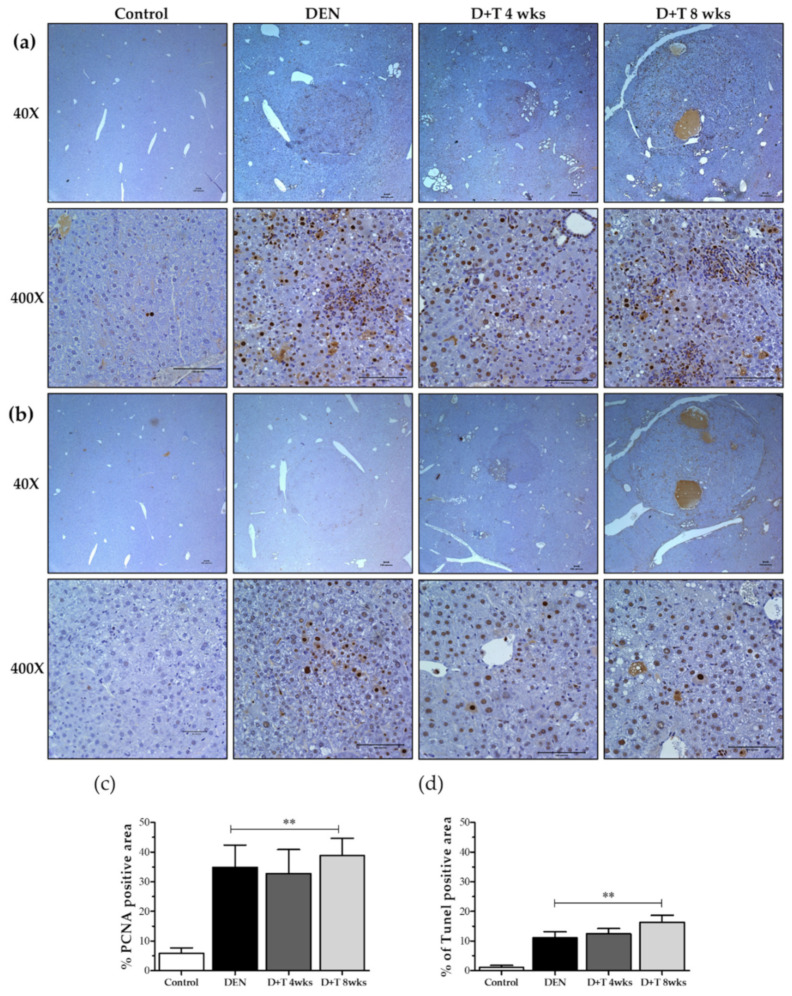
DEN-induced tumors show increased DNA proliferation, DNA damage, and apoptosis. (**a**) Proliferation in DEN induced HCC models shown in Immunohistochemical staining of PCNA as a marker for cell proliferation. Scale bar represents 100 μm. (**b**) Apoptosis in DEN induced HCC models shown as Immunohistochemical staining of TUNEL as a marker for cell death. In all graphs, the values presented are relative to controls. Scale bar represents 100 μm. (**c**) Quantitative analysis for PCNA positive cell. (**d**) Quantitative analysis for TUNEL positive cells. In all graphs, the values presented are relative to controls. Magnification (40× and 400×), values are given as the means ± SD. At least three mice in each group were recorded. ** Significantly different from the control group (*p* < 0.01).

**Figure 4 ijms-21-05461-f004:**
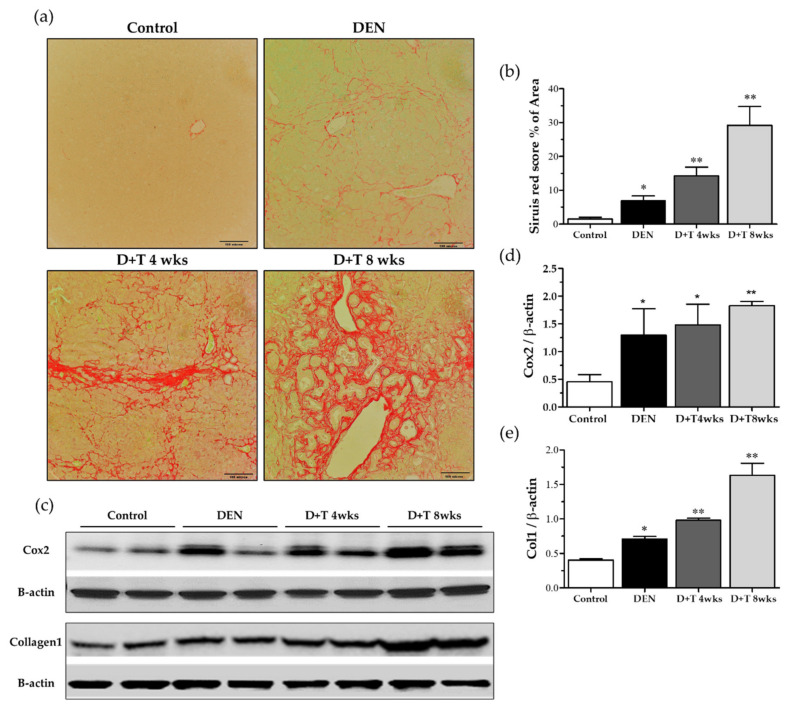
Liver cirrhosis pattern. (**a**) Sirius red staining (200×) for each group. (**b**) Scoring of the Sirius red staining. (**c**) Collagen level and cox2 level by Western blot. β-actin was used as a loading control. (**d**,**e**) Quantitative analysis of collagen and Cox2 normalized with β-actin by the Image J software (NIH). * Significantly different from the control group (*p* < 0.05). ** Significantly different from the control group (*p* < 0.01).

**Table 1 ijms-21-05461-t001:** Body and liver weight in the animal model.

Group	Wks of Treatment	Final Body Weight (g)	Absolute Liver Weight (g)	Relative Liver Weight (g)
Control	0	34.3 ± 1.9	1.08 ± 0.13	3.2 ± 0.49
DEN	8 wks DEN	21.9 ± 3.2 *	1.68 ± 0.20	7.7 ± 2.08 *
DEN+TAA	8 wks DEN + TAA 4 wks	22.7 ± 3.1 *	1.8 ± 0.12	8.2 ± 0.94 *
DEN+TAA	8 wks DEN + TAA 8 wks	22.1±0.6 *	1.8 ± 0.15	8.2 ± 0.58 *

* Significantly different from the control group (*p* < 0.05).

**Table 2 ijms-21-05461-t002:** Liver tumor incidence.

Group	Mice # with Tumor /Total # of Mice	Tumor No.	Tumor No./Animal
Control	0/13	0	0
DEN	29/29	204	7.2 ± 3.42
DEN + TAA 4 weeks	13/13	106	8.81 ± 0.94 ^ns^
DEN + TAA 8 weeks	11/11	126	11.5 ± 2.76 *

* Significantly different from the DEN group (*p* < 0.05), ns (not significant).

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
