# Peer review of "A Modified Protocol of Diethylnitrosamine Administration in Mice to Model Hepatocellular Carcinoma"

_ijms, 2020, doi:10.3390/ijms21155461_

Round 1

Reviewer 1 Report

The study is very interesting and useful.

In the model generation process, the author didn’t show the TAA delivery method, is intravenous or administered intraperitoneally?

About the blood chemistry, did the author test total bilirubin and direct bilirubin? Is there difference between Den alone treated groups?

There is difference on the tumor size between DEN alone and DEN+TAA group, could the author give some explanation?   

Meanwhile, DEN induced HCC is kind of heterogenetic tumor, did the author found the difference characters in different tumor nodules?  Meanwhile, what’s the molecular characters difference between DEN and DEN+TAA induced HCC model, except the fibrosis?

Did the author try the method on other background mice model, will it also works on other background mice?

Did the author use some HCC makers to evaluate the model, such as AFP, or Glutathione S-transferase placental type (GST-P)?  

Did the author monitor the HCC development process in the early and later stage, from the liver weight to body weight, the control group liver weight is a little lower, and the liver weight in the tumor group is not highat autopsy time point, too ?  generally, the normal mouse liver weight to body weight is around 5% . 

Reviewer 2 Report

To be sure that the model represents HCC, it would be nice to have higher magnification (x400)  images with GPC3, vascular and ductular reactions as well as with trabeculae.

Reviewer 3 Report

The manuscript presented by Azra Memon et al. introduced a new HCC mouse model induced by DEN and TAA treatment, which could be of particular interest to the related researchers. However, due to short of introduction and lack of mechanisms, it cannot be published on IJMS.

-The main problem with this manuscript is the reason why the authors chose a high dosage of DEN and the combination of TAA. The standard DEN induced HCC model is single or repeated administration of low dosage over a long period. It is quite confusing the criteria the author chose the DEN concentration from the introduction part. Also, why the author chose TAA as a combination is confusing too. The mechanisms of DEN and TAA to induce HCC are different. What kind of human HCC does the author want to generate from their mouse model? The authors must give enough reason and references in the introduction part.

-The other problem is that the high dosage of two carcinogens raise safety concerns. Although the authors evaluated the survival rate of the mice six weeks after the last dosage, it is not long enough to claim the “high survival rate.” The author should evaluate the survival rate for at least half a year. Also, DEN would induce lung adenoma and leukemia, even at a low dosage. The tumorigenesis in the other organs should be evaluated.

-Figure 2A, to claim that “…Liver nodules presented as basophilic foci with crowded nuclei and were classified as a typical HCC…”, HE stainings at bigger magnification should be provided

-Figure 3A, both of the apoptosis and proliferation co-existed in the liver nodule, which cells are apoptotic and which are proliferating?

-Figure 4A, It is common that TAA could increase liver fibrosis since it would cause oxidative stress and periportal infiltrates. The standard dosage for TAA induced liver fibrosis and HCC is 100-200mg/kg. The dosage used in this model is 300mg/kg, which would cause acute liver failure. Liver function should be evaluated in this DEN+TAA HCC mouse model.

Round 2

Reviewer 2 Report

The main problem in this study is insufficient data for HCC proof.

Authors only mentioned in the Introduction section that DEN provokes GPC3 and AFP expression but did not prove that this happens in their model. Please, provide data.

As I have mentioned in the first review, the larger magnification (x400) of histological images is needed to show readers that your model reflexes HCC histology.

One technical mistake appeared in line 77, looks like a text is missing. Please, correct.

Reviewer 3 Report

The author addressed most of my concerns including the reason why they used DEN and TAA at the same time in induction of HCC. But the safety concern still remains. The authors should address it at least in the discussion part.

Round 3

Reviewer 2 Report

Look through the draft carefuly and correct the mistakes. For example: in line 174 correct the number of the figure for COX-2 data reference. 
